# Nutritional Profile of Donkey and Horse Meat: Effect of Muscle and Aging Time

**DOI:** 10.3390/ani12060746

**Published:** 2022-03-16

**Authors:** Rosaria Marino, Antonella della Malva, Aristide Maggiolino, Pasquale De Palo, Francesca d’Angelo, Josè Manuel Lorenzo, Agostino Sevi, Marzia Albenzio

**Affiliations:** 1Department of Agriculture, Food, Natural Resources and Engineering (DAFNE), University of Foggia, Via Napoli, 71121 Foggia, Italy; antonella.dellamalva@unifg.it (A.d.M.); francesca.dangelo@unifg.it (F.d.); agostino.sevi@unifg.it (A.S.); marzia.albenzio@unifg.it (M.A.); 2Department of Veterinary Medicine, University of Bari A. Moro, 70010 Valenzano, Italy; aristide.maggiolino@uniba.it (A.M.); pasquale.depalo@uniba.it (P.D.P.); 3Área de Tecnología de los Alimentos, Facultad de Ciencias de Ourense, Universidad de Vigo, 32004 Ourense, Spain; jmlorenzo@ceteca.net; 4Centro Tecnológico de la Carne de Galicia, Parque Tecnológico de Galicia, 32900 Ourense, Spain

**Keywords:** meat, donkey, horse, nutritional quality, aging time

## Abstract

**Simple Summary:**

Horse and donkey meat are a good source of nutrients and minerals. However, equine meat in many countries is still unpopular due to its toughness, although it has been demonstrated that an appropriate aging time improves the tenderness in different horse muscles. The present paper investigates the effect of aging time on nutritional properties in different muscles of horse and donkey meat. This information could represent an opportunity to valorize equine animal biodiversity and could be useful for the equine meat industry. The results highlight that horse and donkey meat, being particularly rich in PUFA and EAA, could represent healthy alternatives to traditionally consumed red meat. Aging time did not affect the nutritional profile of horse meat, whereas in donkey meat, a decrease of PUFA was observed after an aging time of 14 days.

**Abstract:**

The aim of this study was to evaluate the effect of 14-day vacuum aging on the nutritional composition of donkey and horse meat. *Longissimus Dorsi* (LD), *Semimebranosus* (SM), and *Rectus Femoris* (RF), *Semitendinosus* (ST) muscles were sampled from each left carcass of 12 donkeys and 12 horses, respectively. Each muscle was divided into three sections, vacuum packaged, and stored at 2 °C for different aging times (1, 6, and 14 days). Fatty acids, amino acids, and cholesterol content were determined. SM exhibited higher levels of polyunsaturated fatty acids (PUFA) both in donkey and horse, whereas LD evidenced higher saturated (SFA) and monounsaturated (MUFA) fatty acids and lower cholesterol content in horse after 1, 6, and 14 days of storage. An aging effect was found only in donkey meat with higher saturated fatty acids and lower PUFA content at the end of the aging period. The highest value of essential amino acids has been displayed in SM an LD muscles from horse and donkey, respectively. Our results highlighted that equine meat, due to an excellent nutritional profile, represents a healthy alternative to traditionally consumed red meat. A different aging method could be used in donkey meat to preserve the high PUFA content.

## 1. Introduction

In recent years, there has been an increased interest of consumers towards meat with a healthy nutritional profile that is environmentally friendly. In this context, equines represent a “sustainable” source of high-quality meat [1] because of its high nutritional value, mainly due to the large use of pastures in the equids rearing system [2,3]. The ratio of polyunsaturated fatty acids n3/n6 in the meat of animals reared in pastures is, in fact, particularly advantageous from a health perspective compared to meat from animals reared in intensive systems. A dietary intake of a lower n6 to n3 ratio is strongly recommended, as a higher intake of n6 fatty acids may reduce the formation of anti-inflammatory mediators from n3 fatty acids [4]. Equidae, being non-ruminant herbivores and hindgut fermenters, can efficiently digest and absorb into the foregut the lipids introduced with the diet before they can reach microbial metabolism in the hindgut. Several studies recognized horse and donkey meat as an excellent source of nutrients, characterized also by a good fatty acid profile rich in omega 3 [5,6,7]. Muscle to muscle variations in chemical composition exist, giving rise to differences among different commercial cuts within the same animal. Understanding the extent of these differences can be useful for enhancing the meat of the whole equine carcass. In addition, our previous studies [8,9] highlighted that aging time affects the tenderness of horse meat, showing different tenderization rate among muscles. No data were available in the literature about the effect of aging time on donkey and horse meat nutritional profiles.

Therefore, the aim of the present study was to evaluate the effect of muscle and aging time (1, 6, and 14 days) on the fatty acids profile, amino acids composition, and cholesterol content of horse and donkey meat. Moreover, in order to deeply characterize equine meat, the comparison between the nutritional profiles of horse and donkey meat are also reported.

## 2. Materials and Methods

### 2.1. Animals, Slaughter Procedure, and Sampling

The study was performed on 12 Martina Franca donkey male foals and 12 Italian heavy draft horse (IHDH) male foals, slaughtered at 12 months of age. A total of 72 muscles were purchased from a local slaughterhouse. Horse muscles (*n* = 36) were collected from *Longissimus dorsi* (LD), *Semitendinosus* (ST), and *Semimembranosus* (SM), whereas donkeys muscles (*n* = 36) were collected from *Longissimus dorsi* (LD), *Rectus femoris* (RF), and *Semimembranosus* (SM), respectively. At approximately 12 months of age, the horses and donkeys were slaughtered according to industrial routines used in Italy and EU rule n. 1099/2009. Carcasses were chilled at 2–4 °C for 24 h, according to standard commercial practices. After 24 h post-mortem, muscles were removed from the left side of each cold carcass (2–4 °C) and then transported under refrigerating condition (at a maximum temperature of 4 °C) to the laboratory. Subsequently, each muscle was divided into three sections, stored at 2 °C under vacuum packaging, and randomly assigned to one of the different aging times (1, 6, and 14 days of aging); cranial and caudal sections were randomized across aging time. Vacuum packaging was performed according to Tateo et al. [10]. Lipid content, fatty acids profile, amino acids composition, and cholesterol content were estimated at each aging time.

### 2.2. Total Lipids Content and Fatty Acids Profile

Analysis of total lipids content was performed, in duplicate, according to AOAC methods (1995) [11]. Fatty acids were extracted according to O’Fallon et al. [12], with slight modifications. Briefly, 1 g of sample was weighed into a screw cap reaction tube, then 0.7 mL of 10 N KOH in water, 5.3 mL of MeOH, and 0.5 mg of C13:0/mL of the internal standard were added. The tubes were incubated in a water bath at 55 °C for 90 min, with hand shaking for 5 s every 20 min, and, once cooled at room temperature, 580 μL of 24 N H_2_SO_4_ was added. The tubes were incubated again, as previously described. After cooling, 3 mL of hexane was added into each tube and the tubes vortexed for 5 min, and then centrifuged at 500× *g* (Eppendorf 5810 R, Eppendorf AG, Hamburg, Germany) for 5 min at 21 °C. Fatty acids methyl esters (FAME) were transferred into a gas-chromatographic vial, and the fatty acid profile was quantified through an Agilent 6890 N instrument (Agilent Technologies, Santa Clara, CA, USA) equipped with an HP-88 fused-silica capillary column (length 100 m, internal diameter 0.25 mm, film thickness 0.25 μm). Operating conditions were as follows: carrier gas (helium) at a constant flow of 1 mL min^−1^; split-splitless injector at 260 °C; split ratio 1:25; injected sample volume 1 μL; FID detector at 260 °C. The temperature program of the column was: 5 min at 100 °C, then increased to 240 °C (3.5 °C min^−1^) and held for 15 min. The retention time and area of each peak were computed using the 6890 N NETWORK GC system software. Fatty acids were identified by comparing their retention times with the fatty acid methyl standards (FIM-FAME-7-Mix, Matreya LLC, State College, PA, USA), added of C18:1,11t, C18:2 9c,11t, and C18:2 9t,11t (Matreya LLC, State College, PA, USA). Results were expressed as g fatty acids/100 g total fatty acids. Atherogenic (AI) and thrombogenic (TI) indices were calculated according to the formula reported by Ulbricht and Southgate [13]:AI = (C12:0 + 4 × C14:0 + C16:0)/(MUFA + Ʃn6 + Ʃn3);
TI = (C14:0 + C16:0 + C18:0)/[(0.5 × MUFA + 0.5 × Ʃn6 + 3 × Ʃn3 + Ʃn3/Ʃn6)].

### 2.3. Amino Acids Determination

Amino acids extraction was carried out according to Marino et al. [14]. Briefly, 20 mg of freeze-dried samples were placed in hydrolysis tubes with 500 μL of 6 M HCl. Tubes were sealed under vacuum and placed in a ventilated oven at 160 °C for 75 min. Hydrolysed samples were filtered through Whatman 0.45 μm filters, and filtered solutions were diluted 1:10 with ultrapure water before being submitted to automated online derivatization and injection. The HPLC system consisted of an Agilent 1260 Infinity Series chromatograph (Agilent Technologies, Santa Clara, CA, USA), equipped with a binary pump (G1312B), a diode-array detector (1315C), and a fluorescence detector (G1321B). The analyses were performed using a Zorbax Eclipse AAA column (150 × 4.6 mm i.d., 3.5 μm particles; Agilent Technologies, Palo Alto, CA, USA). Individual amino acids peaks were identified by comparing their retention times with those of standards. Results for amino acids were expressed as mg/100 g meat.

### 2.4. Cholesterol Determination

Cholesterol content was determined using a quantitative colorimetric kit (BioVision, Waltham, MA, USA) with a UV–Vis spectrophotometer (Biotek PowerWaveXS2, Biotek Instruments, Inc., Highland Park, Winooski, VT, USA). The cholesterol content was expressed as mg/100 g meat.

### 2.5. Statistical Analysis

All the data were subjected to analysis of variance (two-way ANOVA) using the GLM procedure of the SAS statistical software version 9.4 (Cary, NC, USA) [15]. The mathematical model included the fixed effect due to muscle, aging, and muscle × aging, and random residual error.

Results are presented as the least squares means of the data for each muscle, and the variability of the data is expressed by the standard error of the mean (SEM).

All effects were tested for statistical significance (*p* < 0.05), and significant effects were reported in tables. When significant effects were found (*p* < 0.05), Fisher’s LSD test was used to locate significant differences between means. Significant interactions between muscle and aging time were not recorded, so they were not included in tables.

Two-way ANOVA was also performed to test the animal species effect. The model included the fixed effects due to animal species, aging, their interaction, and random residual error. All effects were tested for statistical significance (*p* < 0.05), and significant effects were reported in tables.

## 3. Results

### 3.1. Fatty Acid Profile and Cholesterol Content

The effects of muscle and aging time on lipid content and fatty acids composition of meat from horse and donkey are shown in Table 1 and Table 2, respectively. The type of muscle significantly accounted for the fatty acids composition in both donkey and horse meat. As reported in Table 1, horse LD muscle showed higher content of total lipids (*p* < 0.001) and saturated fatty acids (SFA, *p* < 0.01), with the highest content of miristic (C14:0, *p* < 0.001) and palmitic acids (C16:0, *p* < 0.001), higher monounsaturated fatty acids (MUFA, *p* < 0.01), and lower value of polyunsaturated fatty acids (PUFA, *p* < 0.001), compared to SM and ST. In contrast, the SM muscle showed the highest percentage of PUFA (*p* < 0.001) with the highest value of n6 (*p* < 0.001) and total CLA (*p* < 0.01). Referring to nutritional indices, LD muscle showed lower PUFA/SFA (*p* < 0.01) and n6/n3 (*p* < 0.001), compared to SM and ST.

In donkey meat (Table 2) LD muscle showed a higher percentage of MUFA (*p* < 0.001) and lower content of PUFA (*p* < 0.05), compared to RF and SM, with the highest value of oleic acid (*p* < 0.01) and the lowest value of n6 (*p* < 0.05). As a consequence, LD muscle showed lower PUFA/SFA (*p* < 0.05) compared to RF and SM muscles.

A significant effect of aging time was found only in donkey meat. From 1 to 14 days of aging, SFA increased (*p* < 0.01) due to an increase of stearic acid (C18:0, *p* < 0.01), and PUFA decreased (*p* < 0.05). In particular, a decrease was observed during aging time of both n6 (*p* < 0.05) and n3 (*p* < 0.05) polyunsaturated fatty acids. Consequently, a decrease of PUFA/SFA and an increase of n6/n3, passing from 6 to 14 days, was observed.

The comparison between the fatty acids profile of meat from horse and donkey is shown in Table 3. Horse meat showed higher content of total lipids (*p* < 0.01), saturated fatty acids (*p* < 0.01), with the highest content of miristic, palmitic, and stearic acids (*p* < 0.001), higher monounsaturated fatty acids (MUFA, *p* < 0.001) and lower values of polyunsaturated fatty acids (PUFA, *p* < 0.001), compared to donkey meat. For nutritional indices, horse meat showed lower PUFA/SFA (*p* < 0.01) and n6/n3 (*p* < 0.001) and higher AI (*p* < 0.05) and TI (*p* < 0.05), compared to donkey meat.

An animal × aging effect was found, particularly, in donkey meat, which showed higher stearic acid (*p* < 0.05), SFA (*p* < 0.05), n6/n3 (*p* < 0.01), TI (*p* < 0.05) and lower arachidonic acid (*p* < 0.05), and PUFA/SFA (*p* < 0.05) at 14 days of aging, compared to 1 and 6 days.

Figure 1 shows cholesterol content in horse and donkey meat as affected by muscle and aging time. A significant effect of muscle (*p* < 0.01) was observed only in horse meat. SM muscle showed higher cholesterol content compared to LD and ST muscles. No significant differences on aging effect were found in either equine meat. Comparison between species highlighted that donkey meat showed higher cholesterol content (*p* < 0.05) than horse meat (data not shown).

### 3.2. Amino Acid Composition

The effects of muscle and aging time on the amino acid composition of meat from horse and donkey are shown in Table 4 and Table 5, respectively.

In horse meat, ST muscles showed lower values of aspartate (*p* < 0.05), glutamate (*p* < 0.01), serine (*p* < 0.05), essential amino acids (*p* < 0.05), non-essential amino acids (*p* < 0.05), and total amino acids (*p* < 0.05), compared to LD and SM muscles (Table 4).

An effect of aging time was found only in horse meat, with an increase of histidine (*p* < 0.05) and tyrosine amino acids (*p* < 0.05), progressing from 1 to 6 days of aging.

In donkey meat, *Longissimus dorsi* muscle showed higher values of aspartate, (*p* < 0.05), methionine (*p* < 0.01), isoleucine (*p* < 0.05), lysine (*p* < 0.05), essential amino acids (*p* < 0.05), non-essential amino acids (*p* < 0.05), and total amino acids (*p* < 0.05), compared to RF and SM muscles. No significant aging effect was found.

The comparison between the amino acid composition of meat from horse and donkey is shown in Table 6. Horse meat showed lower content of glutamate (*p* < 0.05), methionine (*p* < 0.01), isoleucine (*p* < 0.05), and leucine (*p* < 0.05), and higher content of proline (*p* < 0.05), histidine (*p* < 0.01), lysine (*p* < 0.001), and essential amino acids (*p* < 0.05), compared to donkey meat. In addition, in horse meat histidine content showed a gradual increase during aging (*p* < 0.05).

## 4. Discussion

The effect of muscle type on fatty acids profile is sparely documented, particularly for donkey meat, and it is controversial. The differences in fatty acid profile among muscles could be attributed to differences in phospholipid concentration, which is greater in red oxidative muscle fiber, compared to glycolytic muscle fiber [16]; therefore, the relatively white LD is generally lower in PUFA percentage than SM. Our results are in agreement with Franco et al. [17] and in disagreement with Tateo et al. [18] and Polidori et al. [19], who did not find significant differences among muscles in horse and donkey meat, respectively.

The different fatty acids profile found in equine muscles partially influenced the health lipid indices. Even if PUFA/SFA in horse and donkey meat is less favorable in *Longissimus Dorsi* muscle compared to the other muscles, it is remarkable, from a nutritional point of view, the lowest n6/n3 found in horse LD muscle. This value is close to the value recommended by World Health Organization (WHO), which should not exceed 4.0, because it is associated with the onset of atherosclerosis and cardiovascular problems [20]. Conversely, in donkey meat, n6/n3 values were slightly higher than the threshold recommended by WHO. This result is linked to the higher linoleic (C18:2) and arachidonic (C20:4) acids content found in donkey meat, compared to horse meat.

However, in the present study, atherogenic and thrombogenic indices are similar among muscles both in horse and donkey meat and are comparable with values found in previous research [21,22]. This result highlights that, in evaluating the nutritional impact of meat, it is important to consider not only the PUFA/SFA and n6/n3 ratios, but also the different metabolic effects of some specific saturated and polyunsaturated fatty acids. Fatty acids can have a very different effect on preventing or promoting atherosclerotic and thrombotic phenomena. The formula of atherogenic and thrombogenic indices highlights that C12:0, C14:0, and C16:0 are atherogenic and C14:0, C16:0, and C18:0 are thrombogenic. Therefore, atherogenic and thrombogenic indices give an effective indication about the risk of atherosclerosis and the sign of the potential aggregation of blood.

The fatty acids profile of horse meat showed the prevalence of SFA and MUFA (39.46% and 40.08%, respectively), whereas donkey meat showed a greater percentage of SFA and PUFA (37.41% and 32.64%, respectively). These results are in agreement with previous studies [23,24,25], nevertheless, a great variation in the fatty acids profile of horse and donkey meat exists due, especially, to the rearing system and to the age at slaughter [3,26].

The higher content of PUFA found in donkey meat, compared to horse meat, could be a consequence of the lean nature of this specie. Indeed, at low levels of fat, the contribution of phospholipids to the fatty acid profile of meat is proportionately greater, and these are more unsaturated than triacylglycerols, which in turn increase in proportion as total lipid increases.

The effect of aging time on fatty acids profile, found only in donkey meat, could be related to a greater amount of PUFA of this specie that led to a greater exposition to oxidative phenomena occurring during the aging period. Therefore, on the basis of this result, we suggest that an alternative aging method to vacuum aging, such as traditional aging or dry aging, could be utilized for the aging process in donkey meat, in order to preserve the excellent fatty acids profile of this specie.

Another important component of the lipid profile is the cholesterol content. Our findings indicated that cholesterol content in horse meat is lower than in other animal species, such as chicken, mutton, beef, and pork [27]. In addition, it was found in a study investigating the effect of moderate consumption of horse meat on the metabolic profile of men and women that the consumption of horse meat significantly reduced serum levels of total and low-density lipoprotein cholesterol and ameliorated the dietary intake of n3 polyunsaturated fatty acids, improving lipid profile and iron status in these subjects [28].

A source of protein is an essential element of a healthy diet, allowing both growth and maintenance of the many thousands of proteins encoded within the human genome.

Controversial results were previously reported on the effect of muscle type on amino acid composition in horse and donkey meat. Lorenzo et al. [29] found statistically significant differences among muscles, whereas Franco et al. [30] and Polidori et al. [6] observed a constant amino acid composition in different commercial cuts in horse and donkey meat, respectively. In the present study, some differences in amino acid composition among muscles have emerged, although this result did not affect the percentages of EEA/AAT, which were comparable among muscles and highlighted the high nutritional value of equine meat. Both horse and donkey meat showed high essential amino acid compared to total amino acid contents percentage, reaching values from 55.53 to 57.07 in horse meat and from 53.62 to 54.49 in donkey meat. Essential amino acids are basic in the diet, particularly for certain population groups with specific needs, like children, the elderly, and those who are sick.

Particularly, in the present study, the most abundant essential amino acid both in donkey and horse meat was lysine, in agreement with previous findings [31,32]. The second most abundant essential amino acid was arginine in horse meat and leucine in donkey meat. In particular, arginine is a functional amino acid, playing an important role in vascular homeostasis, spermatogenesis, and fetal growth. It is considered a conditional essential amino acid, when endogenous synthesis is not adequate to cover metabolic needs, often occurring during children’s growth, as well as during highly catabolic conditions [17].

The increase of histidine and tyrosine amino acids observed in horse meat during aging time could be due to the characteristics of these amino acids. In particular, these amino acids have been highlighted as highly susceptible to ROS action [33], therefore their increase could be due to the protein oxidation that takes place during aging process. The lack of aging effect in donkey meat could be due to different quantitative amino acids composition of this meat that showed more aromatic amino acids.

## 5. Conclusions

Although some differences among muscles have emerged, in regard to fatty acids profile and amino acids and cholesterol content, equine meat displayed excellent nutritional characteristics. Particularly, SM muscle exhibited the most appreciable acidic profile both in donkey and horse, showing higher PUFA and PUFA/SFA, whereas LD muscle showed lower cholesterol content in both equids and higher EAA in donkey meat. However, the results highlighted that horse and donkey meat, being rich in PUFA and EAA, could represent a healthy alternative to traditionally consumed red meat. Aging time was not detrimental for the nutritional properties of equine meat, so an adequate aging time is essential to acquire a new market share. This information could represent an important opportunity in order to valorize the biodiversity of horse and donkey species and could be useful for local farmers to create their own niche in the marketplace.

## Figures and Tables

**Figure 1 animals-12-00746-f001:**
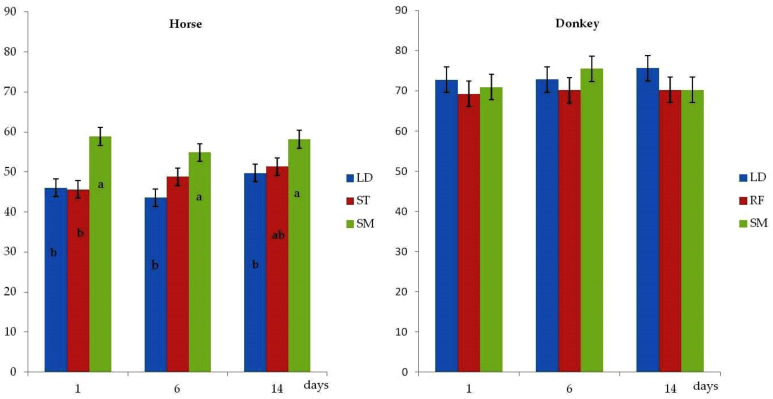
Effect of muscle and aging time (days) on cholesterol content (mg/100 g) of meat from horse and donkey (means ± SEM; LD = Longissimus Dorsi; SM = Semimembranosus; ST = Semitendinosus; RF = Rectus Femoris). a, b = *p* < 0.05 (muscle effect).

**Table 1 animals-12-00746-t001:** Effect of muscle and aging time (days) on total lipids, fatty acid profile (g/100 g fatty acids), and health lipid indices of meat from horse (means ± SEM).

	Aging Time										*p*-Effects	
	1 Day			6 Days			14 Days					
Muscle	LD	SM	ST	LD	SM	ST	LD	SM	ST	SEM	Muscle	Aging
Total lipids	3.21 ^a^	2.28 ^b^	2.03 ^b^	2.98 ^a^	2.17 ^b^	1.95 ^b^	2.85 ^a^	1.95 ^b^	1.75 ^b^	0.26	***	NS
C12:0	0.55	0.57	0.48	0.57	0.59	0.49	0.53	0.55	0.46	0.06	NS	NS
C14:0	4.97 ^a^	4.25 ^b^	4.47 ^b^	4.99 ^a^	4.31 ^b^	4.52 ^b^	4.76 ^a^	4.15 ^b^	4.34 ^b^	0.14	***	NS
C16:0	30.08 ^a^	28.12 ^b^	28.45 ^b^	30.25 ^a^	28.22 ^b^	28.81 ^b^	30.29 ^a^	28.19 ^b^	28.88 ^b^	0.28	***	NS
C18:0	3.91 ^b^	4.65 ^a^	3.88 ^b^	3.78 ^b^	4.38 ^a^	3.66 ^b^	3.75 ^b^	4.51 ^a^	3.98 ^b^	0.16	***	NS
Other SFA	1.04 ^b^	1.44 ^a^	1.42 ^a^	1.02 ^a^	1.55 ^a^	1.28 ^a,b^	1.18 ^a^	1.62 ^a^	1.25 ^a,b^	0.12	*	NS
ΣSFA	40.55 ^a^	39.03 ^b^	38.7 ^b^	40.61 ^a^	39.05 ^b^	38.76 ^b^	40.51 ^a^	39.02 ^b^	38.91 ^b^	0.33	**	NS
C16:1	9.75 ^a^	8.15 ^b^	9.41 ^a^	10.15 ^a^	8.34 ^b^	9.78 ^a^	9.95 ^a^	8.17 ^b^	9.52 ^a^	0.29	***	NS
C18:1t9+t11	0.23	0.28	0.22	0.25	0.27	0.24	0.26	0.25	0.26	0.01	NS	NS
C18:1c9	30.95 ^a^	28.22 ^b^	30.02 ^a^	31.28 ^a^	28.50 ^b^	30.05 ^a^	31.15 ^a^	28.61 ^b^	30.29 ^a^	0.57	***	NS
Other MUFA	0.66	0.67	0.95	0.57	0.58	0.75	0.58	0.55	0.71	0.11	NS	NS
ΣMUFA	41.59 ^a^	37.32 ^b^	40.61 ^a^	42.25 ^a^	37.69 ^b^	40.82 ^a^	41.94 ^a^	37.58 ^b^	40.78 ^a^	0.71	**	NS
C18:2c9c12	11.96 ^c^	16.55 ^a^	14.33 ^b^	11.54 ^c^	16.27 ^a^	14.35 ^b^	11.67 ^c^	16.15 ^a^	14.22 ^b^	0.54	***	NS
CLAc9.t11	0.05	0.05	0.04	0.04	0.05	0.04	0.04	0.06	0.04	0.01	NS	NS
CLAt9.t11	0.04 ^b^	0.15 ^a^	0.09 ^a,b^	0.05 ^b^	0.16 ^a^	0.08 ^b^	0.06 ^b^	0.15 ^a^	0.11 ^a,b^	0.02	*	NS
C20:2n6	0.02	0.04	0.03	0.02	0.04	0.03	0.02	0.04	0.03	0.01	NS	NS
C20:4n6	1.98 ^b^	3.38 ^a^	2.45 ^b^	1.83 ^b^	3.12 ^a^	2.23 ^b^	1.87 ^b^	3.15 ^a^	2.27 ^b^	0.23	***	NS
C22:2n6	0.16	0.21	0.18	0.14	0.22	0.17	0.15	0.2	0.18	0.03	NS	NS
Ʃ n6	14.12 ^c^	20.18 ^a^	17.01 ^b^	13.55 ^c^	19.65 ^a^	16.81 ^b^	13.72 ^c^	19.55 ^a^	16.71 ^b^	0.78	***	NS
TCLA	0.09 ^b^	0.22 ^a^	0.13 ^a,b^	0.09 ^b^	0.21 ^a^	0.12 ^a,b^	0.11 ^b^	0.21 ^a^	0.14 ^a,b^	0.03	*	NS
C18:3n3	3.45	3.05	3.32	3.12	2.71	3.05	3.36	2.88	3.24	0.19	**	NS
C20:3n3	0.04	0.03	0.02	0.04	0.03	0.02	0.04	0.03	0.02	0.01	NS	NS
C20:5n3	0.03	0.05	0.03	0.03	0.05	0.03	0.03	0.05	0.03	0.01	NS	NS
C22:6n3	0.12 ^b^	0.21 ^a^	0.19 ^a,b^	0.12 ^b^	0.22 ^a^	0.18 ^a,b^	0.12 ^b^	0.20 ^a^	0.18 ^a,b^	0.02	**	NS
Ʃ n3	3.64	3.33	3.55	3.31	2.99	3.28	3.55	3.16	3.47	0.21	NS	NS
ΣPUFA	17.85 ^c^	23.71 ^a^	20.67 ^b^	16.93 ^c^	22.85 ^a^	20.18 ^b^	17.36 ^c^	22.91 ^a^	20.31 ^b^	0.82	***	NS
PUFA/SFA	0.44 ^b^	0.61 ^a^	0.53 ^a,b^	0.42 ^b^	0.58 ^a^	0.52 ^a,b^	0.43 ^b^	0.59 ^a^	0.52 ^a,b^	0.04	**	NS
Ʃn6/Ʃn3	3.88 ^c^	6.06 ^a^	4.79 ^b^	4.09 ^c^	6.57 ^a^	5.12 ^b^	3.86 ^c^	6.18 ^a^	4.81 ^b^	0.21	***	NS
AI	0.85	0.75	0.76	0.86	0.76	0.78	0.84	0.75	0.76	0.03	NS	NS
TI	0.95	0.91	0.89	0.98	0.94	0.91	0.96	0.92	0.9	0.03	NS	NS

SFA = saturated fatty acids; MUFA = monounsaturated fatty acids; CLA = conjugated linoleic acid; PUFA = polyunsaturated fatty acids; AI = atherogenic index; TI = thrombogenic index. Other SFA = (C10:0 + C15:0 + C17:0 + C20:0 + C22:0); Other MUFA (C14:1 + C15:1 + C17:1 + C20:1 + C22:1 + C24:1); TCLA = (CLA c9,t11 + CLA t9,t11). AI = (C12:0 + 4 × C14:0 + C16:0)/(MUFA + Ʃn6 + Ʃn3); TI = (C14:0 + C16:0 + C18:0)/[(0.5 × MUFA + 0.5 × Ʃn6 + 3 × Ʃn3 + Ʃn3/Ʃn6)]. Significance: *** (*p* < 0.001), ** (*p* < 0.01), * (*p* < 0.05), and NS = *p* ≥ 0.05. ^a, b, c^ = *p* < 0.05 (muscle effect).

**Table 2 animals-12-00746-t002:** Effect of muscle and aging time (days) on total lipids, fatty acid profile (g/100 g fatty acids), and health lipid indices of meat from donkey (means ± SEM).

	Aging Time										*p*-Effects	
	1 Day			6 Days			14 Days					
Muscle	LD	RF	SM	LD	RF	SM	LD	RF	SM	SEM	Muscle	Aging
Total lipids	2.10	1.78	1.66	1.95	1.63	1.43	1.73	1.55	1.22	0.22	NS	NS
C12:0	0.22	0.26	0.23	0.24	0.28	0.25	0.23	0.27	0.24	0.03	NS	NS
C14:0	3.16	3.36	3.05	3.01	3.15	2.91	2.75	2.98	2.81	0.26	NS	NS
C16:0	26.98 ^a^	24.18 ^b^	24.05 ^b^	27.15 ^a^	24.24 ^b^	24.28 ^b^	27.65 ^a^	24.41 ^b^	24.65 ^b^	0.47	***	NS
C18:0	5.78 ^B^	6.25 ^B^	6.18 ^B^	6.25 ^B^	7.02 ^B^	6.94 ^B^	8.32 ^A^	9.18 ^A^	9.11 ^A^	0.48	NS	**
Other SFA	1.41	1.62	1.68	1.52	1.77	1.76	1.42	1.71	1.75	0.15	NS	NS
ΣSFA	37.55 ^B^	35.67 ^B^	35.19 ^B^	38.17 ^B^	36.46 ^B^	36.14 ^B^	40.37 ^A^	38.55 ^A^	38.56 ^A^	0.55	NS	**
C16:1	5.54 ^a,A^	4.91 ^a,b,A^	4.64 ^b,A^	5.07 ^a,A^	4.51 ^a,b,A^	4.26 ^b,A^	4.62 ^a,B^	4.18 ^b,B^	4.16 ^b,B^	0.19	*	*
C18:1t9+t11	0.38	0.47	0.48	0.35	0.44	0.45	0.31	0.36	0.37	0.04	NS	NS
C18:1c9	26.75 ^a^	22.45 ^b^	21.66 ^b^	26.61 ^a^	22.25 ^b^	21.34 ^b^	27.42 ^a^	23.38 ^b^	22.49 ^b^	0.64	**	NS
Other MUFA	0.76	0.85	0.86	0.77	0.85	0.84	0.73	0.75	0.79	0.03	NS	NS
ΣMUFA	33.43 ^a^	28.68 ^b^	27.64 ^b^	32.81 ^a^	28.04 ^b^	26.89 ^b^	33.08 ^a^	28.67 ^b^	27.81 ^b^	0.91	***	NS
CLAc9.t11	0.12	0.15	0.13	0.11	0.15	0.14	0.13	0.14	0.14	0.02	NS	NS
CLAt9.t11	0.13	0.16	0.18	0.1	0.16	0.16	0.11	0.15	0.17	0.02	NS	NS
C18:2c9.c12	19.76 ^b^	24.05 ^a^	25.05 ^a^	20.11 ^b^	24.36 ^a^	25.47 ^a^	19.37 ^b^	23.75 ^a^	24.66 ^a^	0.62	***	NS
C20:2 n6	0.16 ^A^	0.18 ^A^	0.19 ^A^	0.15 ^A^	0.18 ^A^	0.21 ^A^	0.05 ^B^	0.07 ^B^	0.09 ^B^	0.03	NS	**
C20:4 n6	4.88 ^b,A^	6.21 ^a,A^	7.43 ^a,A^	5.66 ^b,A^	5.98 ^b,A^	7.19 ^a,A^	3.88 ^b,B^	4.35 ^b,B^	5.85 ^a,B^	0.32	*	*
C22:2 n6	0.17 ^A^	0.20 ^A^	0.15 ^A^	0.21 ^A^	0.19 ^A^	0.18 ^A^	0.13 ^B^	0.12 ^B^	0.10 ^B^	0.02	NS	*
Ʃ n6	24.98 ^b,A^	30.65 ^a,A^	32.83 ^a,A^	26.15 ^b,A^	30.73 ^a,A^	33.07 ^a,A^	23.45 ^b,B^	28.31 ^a,B^	30.72 ^a,B^	0.75	*	*
TCLA	0.25	0.31	0.31	0.21	0.31	0.3	0.24	0.29	0.31	0.03	NS	NS
C18:3 n3	2.55 ^b,A^	3.15 ^a,A^	3.22 ^a,A^	2.27 ^b,B^	2.81 ^a,B^	2.82 ^a,B^	2.14 ^b,B^	2.67 ^a,B^	2.58 ^a,B^	0.12	*	*
C20:3 n3	0.21 ^b,A^	0.25 ^a,b,A^	0.32 ^a,A^	0.17 ^b,A^	0.21 ^b,A^	0.33 ^a,A^	0.13 ^b,B^	0.17 ^a,b,B^	0.22 ^a,B^	0.03	NS	*
C20:5 n3	0.35 ^A^	0.31 ^A^	0.32 ^A^	0.29 ^A,B^	0.28 ^A,B^	0.29 ^A,B^	0.22 ^B^	0.21 ^B^	0.24 ^B^	0.03	NS	*
C22:6 n3	0.18	0.17	0.19	0.17	0.18	0.21	0.17	0.17	0.20	0.02	NS	NS
Ʃ n3	3.29 ^b,A^	3.88 ^a,A^	4.05 ^a,A^	2.91 ^b,A,B^	3.48 ^a,A,B^	3.65 ^a,A,B^	2.66 ^b,A^	3.22 ^a,A^	3.24 ^a,A^	0.14	*	*
ΣPUFA	28.52 ^b,A^	34.82 ^a,A^	37.19 ^a,A^	29.26 ^b,A^	34.52 ^a,A^	37.02 ^a,A^	26.35 ^b,A^	31.82 ^a,A^	34.27 ^a,A^	0.82	*	*
PUFA/SFA	0.76 ^b,A^	0.98 ^a,A^	1.06 ^a,A^	0.77 ^b,A^	0.95 ^a,A^	1.03 ^a,A^	0.66 ^b,B^	0.83 ^a,B^	0.89 ^a,B^	0.04	*	*
Ʃn6/Ʃn3	7.63 ^B^	7.95 ^B^	8.15 ^B^	9.07 ^A^	8.88 ^A^	9.11 ^A^	8.87 ^A^	8.85 ^A^	9.53 ^A^	0.19	NS	*
AI	0.64	0.6	0.56	0.64	0.59	0.57	0.65	0.61	0.58	0.03	NS	NS
TI	0.92	0.82	0.78	0.95	0.86	0.83	1.06	0.96	0.94	0.05	NS	NS

SFA = saturated fatty acids; MUFA = monounsaturated fatty acids; CLA = conjugated linoleic acid; PUFA = polyunsaturated fatty acids; AI = atherogenic index; TI = thrombogenic index. Other SFA = (C10:0 + C15:0+C17:0 + C20:0 + C22:0); Other MUFA = (C14:1 + C15:1 + C17:1 + C20:1 + C22:1 + C24:1); TCLA = (CLA c9,t11 + CLA t9,t11); AI = (C12:0 + 4 × C14:0 + C16:0)/(MUFA + Ʃn6 + Ʃn3); TI = (C14:0 + C16:0 + C18:0)/[(0.5 × MUFA + 0.5 × Ʃn6 + 3 × Ʃn3 + Ʃn3/Ʃn6. Significance: *** (*p* < 0.001), ** (*p* < 0.01), * (*p* < 0.05), and NS = *p* ≥ 0.05; ^a, b^ = *p* < 0.05 (muscle effect). ^A, B^ = *p* < 0.05 (aging effect).

**Table 3 animals-12-00746-t003:** Total lipids and fatty acid profile (g/100 g fatty acids) of horse and donkey meat during aging time (means ± SEM).

	Aging Time							*p*-Effects	
	1 Day		6 Days		14 Days				
Animal	Horse	Donkey	Horse	Donkey	Horse	Donkey	SEM	Animal	Animal × Aging
Total lipids	2.51	1.85	2.58	1.67	2.40	1.50	0.25	**	NS
C12:0	0.53	0.24	0.55	0.26	0.51	0.25	0.05	***	NS
C14:0	4.56	3.19	4.61	3.02	4.42	2.85	0.31	***	NS
C16:0	28.88	25.07	29.09	25.22	29.12	25.57	0.45	***	NS
C18:0	4.15	6.07 ^B^	3.94	6.74 ^B^	4.08	8.87^A^	0.32	***	*
Other SFA	1.30	1.57	1.28	1.68	1.35	1.63	0.14	NS	NS
ΣSFA	39.43	36.14 ^B^	39.47	36.92 ^B^	39.48	39.16 ^A^	0.44	**	*
C16:1	9.10	5.03	9.42	4.61	9.21	4.32	0.26	***	NS
C18:1t9+t11	0.24	0.44	0.25	0.41	0.26	0.35	0.03	*	NS
C18:1c9	29.73	23.62	29.94	23.4	30.02	24.43	0.61	***	NS
Other MUFA	0.76	0.82	0.63	0.82	0.61	0.76	0.07	NS	NS
ΣMUFA	39.84	29.92	40.25	29.25	40.1	29.85	0.81	***	NS
C18:2c9c12	14.28	22.95	14.05	23.31	14.01	22.59	0.58	***	NS
CLAc9.t11	0.05	0.16	0.04	0.14	0.05	0.14	0.02	NS	NS
CLAt9.t11	0.09	0.13	0.10	0.13	0.11	0.14	0.02	NS	NS
C20:2n6	0.03	0.07	0.03	0.07	0.03	0.04	0.02	NS	NS
C20:4n6	2.60	6.17 ^A^	2.39	6.28 ^A^	2.43	4.69 ^B^	0.28	***	*
C22:2n6	0.18	0.17	0.18	0.19	0.18	0.12	0.03	NS	NS
Ʃ n6	17.1	29.49	16.67	29.98	16.66	27.49	0.77	***	NS
TCLA	0.14	0.29	0.14	0.27	0.15	0.28	0.03	**	NS
C18:3n3	3.27	2.97	2.96	2.63	3.16	2.46	0.16	**	NS
C20:3n3	0.03	0.26	0.03	0.24	0.03	0.17	0.02	**	NS
C20:5n3	0.04	0.33	0.04	0.29	0.04	0.22	0.02	**	NS
C22:6n3	0.17	0.18	0.17	0.19	0.17	0.18	0.02	NS	NS
Ʃ n3	3.51	3.74	3.19	3.35	3.39	3.04	0.18	NS	NS
ΣPUFA	20.74	33.51 ^A^	19.99	33.61 ^A^	20.19	30.81 ^B^	0.82	***	NS
PUFA/SFA	0.53	0.93 ^A^	0.51	0.91 ^A^	0.51	0.79 ^B^	0.04	**	*
Ʃn6/Ʃn3	4.88	7.88 ^B^	5.22	8.96 ^A^	4.91	9.04 ^A^	0.20	***	**
AI	0.79	0.60	0.80	0.60	0.78	0.61	0.03	*	NS
TI	0.96	0.84 ^B^	0.98	0.88 ^A,B^	0.97	0.98 ^A^	0.03	*	*

SFA = saturated fatty acids; MUFA = monounsaturated fatty acids; CLA = conjugated linoleic acid; PUFA = polyunsaturated fatty acids; AI = atherogenic index; TI = thrombogenic index. Other SFA = (C10:0 + C15:0 + C17:0 + C20:0 + C22:0); Other MUFA = (C14:1 + C15:1 + C17:1 + C20:1 + C22:1 + C24:1); TCLA = (CLA c9,t11 + CLA t9,t11); AI = (C12:0 + 4 × C14:0 + C16:0)/(MUFA + Ʃn6 + Ʃn3); TI = (C14:0 + C16:0 + C18:0)/[(0.5 × MUFA + 0.5 × Ʃn6 + 3 × Ʃn3 + Ʃn3/Ʃn6)]. Significance: *** (*p* < 0.001), ** (*p* < 0.01), * (*p* < 0.05), and NS = *p* ≥ 0.05. ^A, B^ = *p* < 0.05 (animal × aging effect).

**Table 4 animals-12-00746-t004:** Effect of muscle and aging time (days) on amino acids content (mg/100g of meat) of meat from horse (means ± SEM).

	Aging Time										*p*-Effects	
	1 Day			6 Days			14 Days					
Muscle	LD	SM	ST	LD	SM	ST	LD	SM	ST	SEM	Muscle	Aging
Aspartate	1605 ^a,b^	1668 ^a^	1495 ^b^	1602 ^a,b^	1655 ^a^	1496 ^b^	1622 ^a,b^	1678 ^a^	1514 ^b^	38.20	**	NS
Glutamate	2905 ^a^	3055 ^a^	2695 ^b^	2912 ^a^	3061 ^a^	2695 ^b^	2932 ^a^	3062 ^a^	2735 ^b^	65.50	**	NS
Serine	466 ^a^	476 ^a^	418 ^b^	462 ^a^	471 ^a^	413 ^b^	496 ^a^	505 ^a^	446 ^b^	16.10	*	NS
Glycine	655	628	558	688	669	586	678	651	578	36.80	NS	NS
Alanine	807	809	737	817	819	746	796	798	724	39.20	NS	NS
Tyrosine	382 ^B^	401 ^B^	356 ^B^	439 ^A^	461 ^A^	412 ^A^	443 ^A^	464 ^A^	413 ^A^	19.30	NS	*
Proline	888	867	845	876	859	835	886	867	841	36.50	NS	NS
Histidine	795 ^B^	852 ^B^	837 ^B^	928 ^A,B^	985 ^A,B^	972 ^A,B^	1042 ^A^	1095 ^A^	1088 ^A^	57.30	NS	*
Threonine	715	695	638	635	621	561	685	665	605	51.20	NS	NS
Arginine	1318	1321	1188	1295	1302	1168	1319	1325	1180	62.50	NS	NS
Valine	543	587	493	567	605	528	525	566	463	38.50	NS	NS
Methionine	812	809	821	812	809	821	812	809	821	42.30	NS	NS
Phenylalanine	663	671	612	663	671	612	663	671	612	35.40	NS	NS
Isoleucine	700	736	649	700	736	649	700	736	649	35.10	NS	NS
Leucine	1201	1230	1080	1201	1230	1080	1201	1230	1080	62.80	NS	NS
Lysine	3055	3168	2961	3055	3168	2961	3055	3168	2961	73.70	NS	NS
AAT *	17,491 ^a^	17,830 ^a^	16,408 ^b^	17,524 ^a^	18,015 ^a^	16,439 ^b^	17,980 ^a^	18,464 ^a^	16,880 ^b^	298.00	*	NS
EAAT *	9773 ^a,b^	9916 ^a^	9294 ^b^	9732 ^a,b^	10,024 ^a^	9261 ^b^	10,131 ^a,b^	10,443 ^a^	9633 ^b^	235.00	*	NS
NEAAT *	7718 ^a,b^	7914 ^a^	7114 ^b^	7793 ^a,b^	7991 ^a^	7178 ^b^	7848 ^a,b^	8021 ^a^	7246 ^b^	195.00	*	NS
%EA/AAT	55.87	55.61	56.64	55.53	55.64	56.33	56.35	56.56	57.07	0.77	NS	NS

*: AAT = total amino acids; EAAT = essential amino acids total; NEAAT = total non-essential amino acids. Significance: ** (*p* < 0.01), * (*p* < 0.05), and NS = *p* ≥ 0.05^. a, b^ = *p* < 0.05 (muscle effect). ^A, B^ = *p* < 0.05 (aging effect).

**Table 5 animals-12-00746-t005:** Effect of muscle and aging time (days) on amino acids content (mg/100g of meat) of meat from donkey (means ± SEM).

	Aging Time										*p*-Effects	
	1 Day			6 Days			14 Days					
Muscle	LD	RF	SM	LD	RF	SM	LD	RF	SM	SEM	Muscle	Aging
Aspartate	1745 ^a^	1645 ^b^	1658 ^b^	1792 ^a^	1672 ^b^	1658 ^b^	1768 ^a^	1667 ^b^	1658 ^b^	34.50	*	NS
Glutamate	3315	3255	3215	3415	3295	3252	3321.2	3256	3205	48.20	NS	NS
Serine	512	496	495	521	501	495	526	515	494	10.80	NS	NS
Glycine	730	711	705	695	655	667	715	688	685	17.20	NS	NS
Alanine	935	915	906	895	886	873	915	898	879	18.50	NS	NS
Tyrosine	501	479	462	516	488	471	506	484	469	13.80	NS	NS
Proline	568	582	541	571	586	546	584	596	558	16.20	NS	NS
Histidine	438	416	421	445	423	425	432	413	416	12.10	NS	NS
Threonine	1189	1128	1162	1161	1108	1138	1179	1115	1149	28.20	NS	NS
Arginine	1256	1243	1276	1305	1291	1306	1298	1271	1298	37.80	NS	NS
Valine	465	439	438	475	446	444	478	449	451	11.20	NS	NS
Methionine	1119 ^a^	971 ^b^	989 ^b^	1155 ^a^	989 ^b^	1005 ^b^	1138 ^a^	982 ^b^	1015 ^b^	24.50	**	NS
Phenylalanine	628	605	571	648	618	586	658	627	598	21.70	NS	NS
Isoleucine	963 ^a^	875 ^b^	923 ^a,b^	986 ^a^	897 ^b^	945 ^a,b^	944 ^a^	855 ^b^	911 ^a,b^	27.50	*	NS
Leucine	1551	1479	1523	1569	1497	1534	1538	1474	1514	45.50	NS	NS
Lysine	2259 ^a^	2158 ^b^	2141 ^b^	2318 ^a^	2205 ^b^	2195 ^b^	2298 ^a^	2184 ^b^	2158 ^b^	41.30	*	NS
AAT *	18,174 ^a^	17,397 ^b^	17,426 ^b^	18,467 ^a^	17,557 ^b^	17,540 ^b^	18,298 ^a^	17,474 ^b^	17,458 ^b^	111.20	*	NS
EAAT *	9868 ^a^	9314 ^b^	9444 ^b^	10,062 ^a^	9474 ^b^	9578 ^b^	9963 ^a^	9370 ^b^	9510 ^b^	190.00	*	NS
NEAAT *	8306 ^a^	8083 ^a,b^	7982 ^b^	8405 ^a^	8083 ^a,b^	7962 ^b^	8335 ^a^	8104 ^a,b^	7948 ^b^	122.50	*	NS
% EAA/AAT	54.3	53.54	54.19	54.49	53.96	54.61	54.45	53.62	54.47	0.72	NS	NS

*: AAT = total amino acids; EAAT = essential amino acids total; NEAAT = total non-essential amino acids. Significance: ** (*p* < 0.01), * (*p* < 0.05), and NS = *p* ≥ 0.05. ^a, b^ = *p* < 0.05 (muscle effect).

**Table 6 animals-12-00746-t006:** Amino acids composition (mg/100g of meat) of horse and donkey meat during aging time (means ± SEM).

	Aging Time							*p*-Effects	
	1 Day		6 Days		14 Days				
Animal	Horse	Donkey	Horse	Donkey	Horse	Donkey	SEM	Animal	Animal × Aging
Aspartate	1636	1701	1628	1725	1650	1713	36.35	NS	NS
Glutamate	2980	3265	2986	3333	2997	3263	56.85	*	NS
Serine	471	503	466	508	500	510	13.45	NS	NS
Glycine	641	717	678	681	664	700	27.11	NS	NS
Alanine	808	920.5	818	884	797	897	31.15	NS	NS
Tyrosine	391	481	450	493	453	487	22.55	NS	NS
Proline	877	554	867	558	876	571	26.35	*	NS
Histidine	823 ^C^	429	956 ^B^	435	1068 ^A^	424	34.70	**	*
Threonine	705	1175	628	1149	675	1164	39.70	NS	NS
Arginine	1319	1266	1298	1305	1322	1298	50.15	NS	NS
Valine	565	451	586	459	545	464	24.85	NS	NS
Methionine	810	1054	810	1080	810	1076	33.40	**	NS
Phenylalanine	667	599	667	617	667	628	28.55	NS	NS
Isoleucine	718	943	718	965	718	927	31.30	*	NS
Leucine	1216	1537	1216	1551	1216	1526	54.15	*	NS
Lysine	3111	2200	3111	2256	3111	2228	57.50	***	NS
AAT *	17,660	17,800	17,769	18,003	18,222	17,878	204.60	NS	NS
EAAT *	9844	9656	9878	9820	10,287	9736	212.50	*	NS
% EAA/AAT	55.74	54.25	55.59	54.54	56.45	54.46	0.68	NS	NS

*: AAT = total amino acids; EAAT = essential amino acids total. Significance: *** (*p* < 0.001), ** (*p* < 0.01), * (*p* < 0.05), and NS = *p* ≥ 0.05. ^A, B, C^ = *p* < 0.05 (animal × aging effect).

## Data Availability

Data supporting the findings of this study are available from the corresponding author upon reasonable request.

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
