# Peer review of "Nutritional Profile of Donkey and Horse Meat: Effect of Muscle and Aging Time"

_animals, 2022, doi:10.3390/ani12060746_

Round 1

Reviewer 1 Report

The work "Nutritional Profile of Donkey and Horse Meat: Effect of Muscle and Aging Time" has already been reviewed by a reviewer.
The new version of the work did not address all the comments of the reviewer, especially those concerning research methods (e.g. Please provide the method and conditions for packing the meat in a vacuum. Why were 1, 6, 14 days chosen? Why were muscles packed in a vacuum and not in a high-oxygen atmosphere? Justify.)
Changes have been introduced to the work only partially compared to the previous version.
The changes made to the introduction section should be consistent with the rest of the text. I suggest you check this section carefully.
The authors do not use punctuation marks in the fragments marked in red. One can only guess where one sentence ends and another begins.
The text requires native-speaker proofreading.

Author Response

The work "Nutritional Profile of Donkey and Horse Meat: Effect of Muscle and Aging Time" has already been reviewed by a reviewer. The new version of the work did not address all the comments of the reviewer, especially those concerning research methods (e.g. Please provide the method and conditions for packing the meat in a vacuum. Why were 1, 6, 14 days chosen? Why were muscles packed in a vacuum and not in a high-oxygen atmosphere? Justify.)
Changes have been introduced to the work only partially compared to the previous version. The changes made to the introduction section should be consistent with the rest of the text. I suggest you check this section carefully. The authors do not use punctuation marks in the fragments marked in red. One can only guess where one sentence ends and another begins. The text requires native-speaker proofreading.

AU: We would like to thank this reviewer for the comments and useful suggestions that have contributes to improve the quality of the manuscript.

Please provide the method and conditions for packing the meat in a vacuum.

AU: Vacuum meat packing method and conditions have been performed according to Tateo et al (Animals 2020, 10, 1495.https://doi.org/10.3390/ani10091495). Briefly, all samples were vacuum packaged with Besser Vacuum® film (Besser Vacuum, Dignano, Udine, Italy; characterized by 65 µm thickness, 63 g/m2 of weight, ≤ 65 cm3/m2 × day × bar of oxygen permeability at 23 °C and 0% of relative humidity, and ≤ 3.5 g/ m2 × day of water vapor permeability at 23°C and 85% of relative humidity) by using a Tecla Jumbo Inox vacuum device. However, the reference has been added to the revised manuscript.

Why were 1, 6, 14 days chosen? Justify.
AU: Given that the animals involved in our study were young (about 12 months of age), we set the aging period to 14 days (1 day has been chosen as starting point and 6 days as the middle point). The choice of 1 and 6 days, in particular, was done according to meat marketing in the large distribution, because, usually, equine meat is not aged but sold into the market within a few days after slaughter.

Why were muscles packed in a vacuum and not in a high-oxygen atmosphere? Justify.

AU: Muscles were packed in a vacuum and not in a high-oxygen atmosphere in order to extend meat conservation and its marketing in the large distribution without a detrimental effect on eating quality. In fact, previous studies (Geesink et al., 2015 MS,104:85-89; Frank et al 2017 Meat Science, 123, 126–133) have shown that MAP negatively affects eating quality of beef particularly, reducing tenderization. On the contrary, vacuum packaging has been shown to lead to an increase in tenderness and juiciness.

Changes have been introduced to the work only partially compared to the previous version. The changes made to the introduction section should be consistent with the rest of the text. I suggest you check this section carefully.

AU: The introduction section has been checked and revised as suggested by the reviewer.

The authors do not use punctuation marks in the fragments marked in red. One can only guess where one sentence ends and another begins.

AU: Punctuation was checked and added where missing at the end of each sentence.

The text requires native-speaker proofreading.

AU: Editing of the English language has been performed.

Reviewer 2 Report

The authors addressed changes, and with that, the article is in conditions to be accepted for publication.

Some minor suggestions

L44 is in fact particularly "change with" is, in fact, particularly

L56 In addition our "change with" In addition, our

L59  in literature about  "change with" in the literature about

L147 affected significantly the fatty acids composition in both equids "change with" affected the fatty acids composition significantly in both equids.

L256-257 In fact, fatty acids can have a very different effect on the prevention or promotion of atherosclerotic and thrombotic phenomena. "change with" In fact, fatty acids can have a very different effect on preventing or promoting atherosclerotic and thrombotic phenomena.

Author Response

The authors addressed changes, and with that, the article is in conditions to be accepted for publication.

 Some minor suggestions

L44 is in fact particularly "change with" is, in fact, particularly

AU: It has been done

L56 In addition our "change with" In addition, our

AU: It has been done

L59  in literature about  "change with" in the literature about

AU: It has been done

L147 affected significantly the fatty acids composition in both equids "change with" affected the fatty acids composition significantly in both equids.

AU: It has been done

L256-257 In fact, fatty acids can have a very different effect on the prevention or promotion of atherosclerotic and thrombotic phenomena. "change with" In fact, fatty acids can have a very different effect on preventing or promoting atherosclerotic and thrombotic phenomena.

AU: It has been done

Round 2

Reviewer 1 Report

I want to thank the authors for their explanations and changes to the article.

I have one more comment . Line 51. I propose to remove "and minerals". Nutrients are also vitamins and minerals. On the other hand, minerals are not vitamins, and the authors indicate them in parentheses. 

Author Response

I have one more comment . Line 51. I propose to remove "and minerals". Nutrients are also vitamins and minerals. On the other hand, minerals are not vitamins, and the authors indicate them in parentheses. 

AU:  “and mineral” has been removed as suggested by reviewer , all the sentence is highlighted in red.

This manuscript is a resubmission of an earlier submission. The following is a list of the peer review reports and author responses from that submission.

Round 1

Reviewer 1 Report

Interesting topic of the work. However, the manuscript contains serious errors. The work should be thoroughly redrafted. 
Selected detailed comments 

Abstract 
Line 25 - please remove “and LD, SM”. 
Lines: 28-30 please add the period (on which day of storage 1, 6, 14, and maybe in control trials?).

Introduction
 I suggest editing so that the Introduction section is related to the purpose of the work. 
Lines: 39-41 I propose to delete because this fragment is not related to the topic of the work. 
Line 44. The authors report that horse consumption is increasing, while it was not specified where, and the reference is from 2015. 
Lines 49-56 does not apply to the purpose of the work. Please delete.
Lines 59-62 please re-edit this passage. 

Materials and Methods. 
Line 70 should be “amino acids”. 
Lines 130-131. Unclear sentence. 
Please provide the method and conditions for packing the meat in a vacuum. Why were 1, 6, 14 days chosen? Why were muscles packed in a vacuum and not in a high-oxygen atmosphere? Justify. 
Table 1-4 - What does day 0 mean? There is no that day in the research methodology. What about Day 1? 
Tables 1 and 2. For example, is SFA the sum of saturated fatty acids? I suggest you clarify the abbreviation below the table or add the word sum to the SFA. The same is true for the other groups of fatty acids. 
Statistical Analysis - please add the type of analysis of variance used. 
There is also no information about the indicators in the Results section. 
Lines 148, 156 and in other tables - Unify the spelling "p".

Delete table 3 I 4. Transfer data from tables to appropriate tables 1 and 2. 
Figure 1. Explain the symbols T1, T6, T14.

The charts' muscles do not agree with those in the test method (lines 68-71).

Discussion of the results. 
As there are errors in the Results part, it isn't easy to refer to the discussion of the results. Indeed, the discussion should explain the changes in the muscles, including comparing horse and donkey meat. 

Reviewer 2 Report

General comments:

The topic of this article is of high interest. The manuscript contains some important findings to draw conclusions.  However, there are some publications on the nutritional characteristics of donkey or horse meat. And there are plenty of peer-reviewed articles about changes in fatty and amino acids during aging periods. Thus, it is difficult to think that this experiment has originality compared to the previous studies.

Specific comments:

In general, for tough meat, it takes 14 d or more to improve tenderness due to aging. Why did the authors set the aging period to 14 d in this study?

Line 45: delete blank before “Several”.

Line 65-: provide a lipid content method.

Tables and Figures: provide full names for abbreviations.

Line 103: provide the calculation of lipid indices and full names for abbreviations.

Line 134-: Results should be written in the order of items in the table. For example, lipid content first. Otherwise, please revise the order of items.

Line 158: delete “As shown in table 2”.

Line 173: delete “As shown in table 4”.

Line 174: Particularly,

Table 3 and 4: change “nutritional indices” to “nutritional quality indices of the lipids” or “health lipid indices”.

Figure 1: It would be nice if the resolution was a little higher.

Figure 1: T1, T6, T14 à 0 day, 6 days, 14 days (like Tables).

Figure 1: ST to RF. Provide full names.

Line 195: delete “As reported in table 6”.

Line 205: change “muscles” to “muscle fibre”

Line 224: provide the full name of OMS.

Line 237-238: remove or revise this sentence. It cannot be regarded as the result of this experiment.